# Rotavirus Spreads in a Spatially Controlled Manner

**DOI:** 10.3390/cells14040313

**Published:** 2025-02-19

**Authors:** Gianna V. Passarelli, Patricio Doldan, Camila Metz-Zumaran, Yagmur Keser, Steeve Boulant, Megan L. Stanifer

**Affiliations:** 1Department of Molecular Genetics and Microbiology, College of Medicine, University of Florida, Gainesville, FL 32601, USA; 2Department of Infectious Disease: Virology, Heidelberg University Hospital, 69205 Heidelberg, Germany

**Keywords:** rotavirus, virus spread, spatial virus replication

## Abstract

Rotavirus is an enteric virus that leads to 200,000 deaths worldwide every year. The live-cell imaging evaluating rotavirus infection of MA104 cells revealed that rotavirus replication and spread occurs in a spatially controlled manner. Specifically, following initial rotavirus infection, the infected cells die, and the second round of infection occurs in the restricted area surrounding the initially infected cell. Interestingly, we found that the time required to establish the secondary infection is shorter compared to the time required for the initial infection. To determine if this increase in the kinetic of secondary infection was due to the early release of viruses or priming of the cells that are infected during the secondary infection, we used a combination of live-cell microscopy, trypsin neutralization assays, and the pharmacological inhibition of calcium signaling. Together, our results show that the second round of infection required rotavirus to be released and accessible to extracellular proteases. In addition, we found that the calcium wave induced upon rotavirus infection was critical for initial infection but did not play a role in the establishment of a secondary infection. Finally, we uncovered that high viral titers released from the initial infection were sufficient to accelerate the rate of the secondary infection.

## 1. Introduction

Enteric viruses, such as rotavirus, target the intestinal epithelium to initiate their infection [1]. Children and immunocompromised adults worldwide are most at risk with rotavirus infections, potentially leading to severe gastroenteritis, nausea, vomiting, and diarrhea [2]. Due to the diarrhea-induced dehydration of children in developing countries, rotavirus-associated gastroenteritis results in more than 200,000 deaths per year worldwide [2]. Despite the development of multiple vaccines, their efficacy is lower in developing countries compared to developed countries, where rotavirus remains an important public health issue [3,4]. As a result, it is crucial to understand the mechanism of how rotavirus replicates and spreads to design interventions, aiming at better controlling rotavirus infection.

Rotavirus is a non-enveloped segmented double-stranded RNA virus that consists of 11 individual genes [1]. Six of those genes encode for structural proteins (VP1, VP2, VP3, VP4/5/8, VP6, and VP7), while the remaining five genes encode for non-structural proteins (NSP1, NSP2, NSP3, NSP4, NSP5/6), which are involved in viral genome replication and controlling the innate immune response [1]. Rotavirus consists of three layers: an outer, middle, and inner layer [5]. The outermost capsid consists of spike proteins VP4/5/8 and the outer layer of VP7 [5]. Underneath VP7 is the middle layer consisting of VP6 and an innermost layer of VP2, which surrounds VP1/3 and the viral double-stranded RNA genome [5]. Rotavirus utilizes the tail of its spike proteins (VP4) to bind to host receptors [6]. Receptor-mediated endocytosis is then triggered by β1 integrins [7]. In the endosome, the outer layer of the capsid containing VP4/5/8 and VP7 is degraded or removed upon conformational change via trypsin-like proteases [8]. In this subviral state, the virus is now considered active and able to escape the endosome to release its core structure into the cytoplasm and begin transcription [8]. De novo viruses assemble in viroplasms where genome replication occurs [9,10]. Upon particle assembly, the viruses bud at the endoplasmic reticulum and then egress to infect neighboring cells [11]. It has been shown that rotaviruses can shed as non-enveloped virus particles or within microvesicles, which can transmit high levels of infectious virus [12,13].

The NSP4 viral protein is rotavirus’ main tool to disrupt the intestinal epithelium upon infection via its induction of calcium waves [14,15]. Upon infection with rotavirus, NSP4 dysregulates calcium homeostasis by mediating intracellular calcium release from the endoplasmic reticulum (ER) into the cytoplasm. This causes the infected cell to swell, damaging the cytoskeleton, and leads to the extracellular release of ADP [16]. ADP binds the purinergic receptor P2Y1 on both the infected cells and on neighboring cells. ADP binding to P2Y1 induces the activation of the G-protein complex that activates the phospholipase C, which in turn catalyzes the cleavage of phosphatidylinositol 4,5-bisphosphate (PIP2) into the second messengers inositol 1,4,5-trisphosphate (IP3) and diacylglycerol (DAG). IP3 binds to the IP3 receptors (IP3Rs) on the endoplasmic reticulum, triggering calcium release from internal stores [16,17]. This calcium release is often referred to as a “calcium wave” [18] as it propagates from cell to cell in an excentric manner from the primary infected cell. It has been previously shown that rotavirus significantly increases cytosolic calcium during the peak of rotavirus replication [14,16]. Calcium waves have been shown to be a key part of rotavirus replication [19], but whether calcium waves impact rotavirus spread to adjacent cells is still unknown.

We employed live-cell microscopy to evaluate how rotavirus spreads to neighboring cells. Our imaging experiments revealed that, after initial infection, the rotavirus-infected cell dies and releases de novo viruses. Interestingly, our analysis has revealed that (1) rotavirus spread is spatially controlled where only cells directly adjacent to the initially infected cell become infected, and (2) the second round of infection occurred at a faster rate than the initial infection. We determined that this increased rate of infection was not due to encapsidated de novo viruses or to the induced calcium wave. Instead, we discovered that increased virus load released following initial infection led to a greatly shortened rotavirus lifecycle during the second round. Together, these results show that rotavirus spreads in a spatially controlled manner and can increase its replication by increasing virus load during replication.

## 2. Materials and Methods

### 2.1. Cell Culture

MA104 African green monkey kidney epithelial cells (ATCC CRL-2378.1) were maintained in a minimum essential medium (MEM; Fischer, Waltham, MA, USA catalog# MT10010CV) with an addition of 10% fetal bovine serum (FBS), 1% penicillin–streptomycin (Gibco, Waltham, MA, USA), and 2 mM of l-glutamine (Gibco, Walthan, MA, USA). MA104 cells were incubated at 37 °C with 5% CO_2_ and 21% O_2_. MA104 cells were kept in T75 flasks and split 1:10 once confluent using 0.05% Trypsin/EDTA.

### 2.2. Viruses

SA11 simian rotaviruses expressing fluorescent marker NSP3 UnaG were kindly provided by John Patton (Indiana University) [20,21]. All rotaviruses were amplified, produced, and titered in MA104 cells. Briefly, the virus was amplified by adding trypsin activated rotavirus at an MOI of 0.03 to three confluent T175 flasks of MA104 cells. MA104 cells were maintained in serum-free media with 0.5 µg/mL of trypsin. The 72 h post-infection (hpi) cells were lysed by repeated freeze thaw cycles (3×), the virus was semipurified by ultracentrifugation (25,000× *g*, 1.5 h) through a sucrose cushion, and the pellet was resuspended in serum-free phenol-red free OptiMem. Produced virus stocks were titered by plaque assay in MA104 cells, as described below.

### 2.3. Pharmacological Inhibitors

BPTU (Tocris, Bristol, UK catalog# 6078) was used at a final concentration of 10 µM.

### 2.4. Rotavirus Infections

MA104 cells were seeded at 100,000 cells per well in a 48-well plate and incubated for two days at 37 °C prior to infection. Rotavirus stocks were pretreated with 0.5 µg/mL of trypsin (TPCK treated from bovine pancreas, Sigma, St. Louis, MO, USA. Catalog# T1426) in FBS-free media for 30 min at 37 °C. Following virus activation, the cells were washed twice with PBS and then infected with activated rotavirus particles at the MOI indicated in the figure legends. The plates were incubated for 1 h at 37 °C and were rocked every 15 min. The inoculum was aspirated 1 h post-infection, and the cells were washed one time with PBS. Finally, FBS-free media containing 0.5 µg/mL of trypsin were added back onto the cells for the remainder of the experiment. The infection was monitored by live-cell microscopy, and the wells were imaged every 30 min for 48 h in a Zeiss Celldiscoverer 7 microscope.

### 2.5. Rotavirus Infection with Trypsin Addition Post-Infection

MA104 cells were seeded at 100,000 cells per well in a 48-well plate and incubated for two days at 37 °C prior to infection. Rotaviruses were pretreated with 0.5 µg/mL of trypsin (TPCK treated from bovine pancreas) in FBS-free media for 30 min at 37 °C. Following virus activation, the cells were washed twice with PBS and then infected with activated rotavirus particles at the MOI indicated in the figure legends. The plates were incubated for 1 h at 37 °C and were rocked every 15 min. Following 1 h of infection, the inoculum was aspirated, and the cells were washed one time with PBS. Following virus removal, the media was replaced with FBS-free media without trypsin. Cells were then imaged by live-cell microscopy every 30 min for 48 h using a Zeiss Celldiscoverer 7 microscope. At 10, 12, 14, and 16 h post-infection (hpi), the imaging was paused at each time point, and FBS-free media containing 0.5 µg/mL of trypsin was added. Imaging was then restarted and continued through the remainder of the experiment.

### 2.6. Rotavirus Infection with Trypsin Inactivation Using FBS

MA104 cells were seeded at 100,000 cells per well in a 48-well plate and incubated for two days at 37 °C prior to infection. Rotaviruses were pretreated with 0.5 µg/mL of trypsin (TPCK treated from bovine pancreas) in FBS-free media for 30 min at 37 °C. Following virus activation, the cells were washed twice with PBS and then infected with activated rotavirus particles at the MOI indicated in the figure legends. The plates were incubated for 1 h at 37 °C and were rocked every 15 min. Following 1 h of infection, the inoculum was aspirated, and the cells were washed one time with PBS. The media were replaced with a fresh concentration of 0.5 µg/mL of trypsin and left to incubate at 37 °C. Cells were then imaged by live-cell microscopy every 30 min for 48 h using a Zeiss Celldiscoverer 7 microscope. At 6, 8, 10, 12 hpi, the imaging was paused at each time point, and FBS was added to each well to a final concentration of 10% to inactivate trypsin. Imaging was then restarted and continued through the remainder of the experiment.

### 2.7. Live-Cell Fluorescence Microscopy and Image Analysis

Cells were infected as described above, and viral infections were imaged over time using a Celldiscoverer7 (Zeiss, Oberkochen, Germany). During experiments, cells were kept at 37 °C and 5% CO_2_. A 5× magnification objective with a 1× magnification lens was used. For rotavirus infection and spread, UnaG was imaged with a 470 nm LED lamp at 50% laser power and 300 ms exposure time. The brightfield image was acquired in TL Phase Gradient at 10% laser power and 1 ms exposure time. The Focus Strategy was set to Definite Focus, and Acquisition Mode was set to 2 × 2 binning. The Tiles option allowed for multiple images per well to be taken in various spots, and 25 tiles were taken per field of view. Images were acquired at 30 min intervals for 16 to 24 h post-infection. Data were analyzed using ImageJ Fiji (2.9.0 version). The number of infected cells was counted using a mask in ImageJ to determine the number of nuclei in a field of view. The number of rotavirus positive cells (the number of UnaG positive cells) per field of view was manually counted, and the percentage of infection was determined by the ratio of UnaG positive cells to nuclei. The time to initial infection, time to cell death, and time to colony formation were determined by manually following single infected cells throughout the 36-to-48 h movie.

### 2.8. Plaque Assays

MA104 cells were seeded in 6-well plates with 300,000 cells per well and incubated between 6 and 10 days to attain a confluent monolayer of cells. Virus supernatant and cell lysates were activated by 0.5 µg/mL of trypsin (TPCK treated from bovine pancreas) in FBS-free media for 30 min at 37 °C. Following activation, serial dilutions ranging from 10^−1^ to 10^−8^ were prepared. Cells were washed twice with FBS-free MEM media, and 500 µL of the diluted virus was added onto the cells and incubated for one hour at 37 °C. The plates were rocked every 15 min to allow for a homogenous infection across the MA104 monolayer. After 1 h of virus incubation, cells were washed with an FBS-free MEM and overlayed with an MEM medium containing 0.5 µg/mL of trypsin and 0.6% agarose. Plaque assays were incubated for 3 to 7 days at 37 °C. Once plaques had formed, Neutral Red Solution (0.33%) (Sigma-Aldrich, St. Louis, MO, USA N2889) was diluted in an FBS-free 2X MEM (2X MEM, Thermo, 11935046) in a 1:20 ratio, and 1 mL of the diluted Neutral Red Solution was added to each well. Neutral Red Solution was incubated on the overlay for 4–5 h at 37 °C.

### 2.9. Statistical Analysis and Generation of Schematics

All graphs were created in GraphPad Prism (version 10.4.1) and *t*-tests were performed as described in the figures. All diagrams were generated using BioRender (version 4.2.2) (Toronto, ON, Canada, https://www.biorender.com/, accessed on 29 December 2024).

## 3. Results

### 3.1. Rotavirus Spreads in a Spatially Controlled Manner

While studying the importance of type III interferon in controlling rotavirus infection, we observed that infection was spatially restricted with rotavirus infecting clusters of cells (Figure 3 of [22]). To evaluate the underlying mechanisms by which rotavirus infection leads to the formation of colonies of infected cells, we employed live-cell microscopy to follow rotavirus infection and spread. MA104 cells were infected with wild-type SA11 rotavirus expressing the fluorescent protein UnaG [20,21]. Infection was performed at the low multiplicity of infection (MOI) (0.012, 0.003, and 0.0012) to allow for monitoring the spread of rotavirus infection from infected to non-infected cells. The fluorescent protein UnaG is only expressed upon virus replication and serves as a marker of virus infection [21,22]. Rotavirus infection was imaged every 30 min for 16 h. Time course imaging revealed that rotavirus infection was first limited to a few numbers of cells, which correlated with the low MOI use for infection. Viral infection (detection of UnaG signal) was first detectable around 8 h of post-infection (hpi) (Appendix A and Figure 1A–C). We observed that the fluorescence intensity of these infected cells increased overtime as rotavirus replicated (Appendix A). Interestingly, 5–6 h after detecting the UnaG protein in the primary infected cells (around 12 hpi), we observed cell membrane ruffling and cell death (Appendix A and Figure 1A,D). Following the death of the primary infected cells, we observed a second round of infection in a spatially restricted manner with the newly infected cells forming a colony surrounding the initial primary infected cells (Appendix A and Figure 1A,B). Interestingly, while primary infection was on average detected at 8 hpi (Figure 1C), the second round of infection occurred more rapidly and was visualized by 2–4 h post-death of the initially infected cells (Figure 1E). Together, these data show that rotavirus infection spreads in a spatially controlled manner where one initial infection leads to the formation of a colony of infected cells during subsequent rounds of infection.

### 3.2. Progeny Virions Are Released and Are Exposed to the Extracellular Environment

Detection of UnaG in the primary infected cells was evident at 8 hpi (Figure 1C). However, UnaG was detectable in the secondary infected cells as early as 3 h after death of the primary infected cells (Figure 1E). These results suggest that the second round of infection occurs at faster kinetics compared to the initial infection (Figure 1C,E). This difference in the time required to detect the UnaG expression between the primary and secondary round of infection could be due to a priming of cells surrounding the primary infected cells, which speeds up the second round of infection or could be the result of de novo infectious rotavirus particles being released prior to cell death of the primary infected cells. To evaluate when de novo infectious rotavirus particles are released from the primary infected cells, MA104 cells were infected with wild-type SA11 rotavirus expressing UnaG. At 4, 6, 8, 10, and 12 hpi, supernatants were harvested and analyzed by plaque assay. Results show that infectious virus particles are released from the primary infected cells at 12 hpi (Figure 2A), which coincides with the time of cell death, which was 4 h post-detection of the UnaG signal that was detectable at 8 hpi (Figure 1D and Figure 2A). This observation suggests that viruses are not released prior to the cell death of the primary infected cells and, therefore, suggest that the faster kinetics of the second round of infection is not due to infectious virus particles being released before the death of the primary infected cells.

When released from cells, rotavirus particles are not infectious unless they are proteolytically activated [1]. During infection of the gastrointestinal tract, this is mediated by the serine proteases present in the gut lumen. In an in vitro infection model, the rotavirus must be activated by trypsin to infect cells, and trypsin must be present in the cell culture medium for rotavirus infection to spread. However, previous reports have suggested that rotaviruses are not released as individual non-enveloped particles but can be released in microvesicles, which then deliver high quantities of viruses to the subsequent cells [12]. When rotavirus is packaged in microvesicles, it is activated and does not require trypsin activation. In addition, viruses packaged in microvesicles can enter cells in a receptor-independent manner [12]. To determine if the secondary round of rotavirus infection is mediated by viruses packaged in microvesicles or if viruses are released as unenveloped particles requiring trypsin activation, MA104 cells were infected with activated wild-type SA11 rotavirus expressing UnaG. Following a 1 h internalization, unbound viruses were removed through extensive washing, and infection was allowed to proceed in the absence of trypsin. Trypsin was then added to the cell culture medium at 1, 10, 12, 14, and 16 hpi (Figure 2B). Virus infection was visualized by live-cell microscopy, acquiring an image every 30 min for 18 h (Figure 2C). Results show that, when trypsin was added at 1 and 10 hpi, rotavirus infection proceeded as before, with secondary rounds of infection occurring between 14 and 18 hpi (Figure 2C,F,G). However, when trypsin was added at 12, 14, and 16 hpi, the onset of the second round of infection was delayed (Figure 2C,D,H). Importantly, delaying the start of secondary infection did not affect the number of primary infected cells that led to the formation of rotavirus-infected cellular colonies during the secondary round of infection (Figure 2E).

To further confirm that rotaviruses are released as non-enveloped particles and not as sealed within a vesicle, we performed a trypsin neutralization assay. Following binding and internalization, the unbound rotavirus was removed, and fresh media containing trypsin was added to the cells to allow for secondary rounds of rotavirus infection (Figure 3A). To confirm that virus was released at 12 hpi and required extracellular trypsin for subsequent infections, FBS was added to the wells to inactivate trypsin activity at 6, 8, 10, and 12 hpi (Figure 3A). Results show that, when FBS was added prior to 12 hpi, there were no secondary rounds of infection that occurred, and only primary infections were detected (Figure 3B–D). However, when FBS was added at 12 hpi, secondary rounds of infection were detectable. Importantly, the quantification of the number of rotavirus-infected cellular colonies revealed that the inhibition of trypsin 12 hpi did not impact the number of primary infected cells that led to the formation of rotavirus-infected cellular colonies during the secondary round of infection compared to the control condition (Figure 3B–D). Together, these data show that rotavirus is released from cells during cell death, and these viruses require extracellular trypsin to initiate subsequent rounds of infection.

### 3.3. Calcium Signaling Is Required for Rotavirus Colony Formation

Our data have shown that rotavirus infection spreads in a spatially controlled manner and that the second round of infection occurs with increased kinetics as compared to the first round of infection (Figure 1). We have determined that this priming of the second round of infection is not due to the virus being released prior to cell death (Figure 2A) or spread within enclosed vesicles as the rotavirus required exposure to extracellular trypsin for the second round of infection (Figure 2 and Figure 3). Rotavirus has been shown to induce calcium waves through the secretion of ADP, which binds to the P2Y1 receptor [16,17]. These calcium waves are important for its spread [19]. To determine whether calcium waves induced by the primary infection helped to prime the secondary infection to occur with increased kinetics, we employed the P2Y1 antagonist BPTU [23]. MA104 cells were infected with rotavirus. Cells were either mock-treated or treated with BPTU at the time of infection, at 4 hpi, or at 8 hpi. Rotavirus infection was monitored by live-cell microscopy every 30 min for 24 h. Results show that, when BPTU is added at the time of infection (0 hpi) or at 4 hpi, rotavirus infection is strongly impaired. Blocking P2Y1 signaling, through BPTU treatment, early in rotavirus infection leads to a decrease in primary infection (Figure 4A,B). Importantly, while, in mock-treated cells, we observed the formation of rotavirus-infected cellular colonies following the death of the primary infected cells (Figure 1 and Figure 4A,C and Appendix A), we failed to observed these colonies when BPTU was added at 0 or 4 hpi (Figure 4A,C,F and Appendix A). Interestingly, while the time to initial infection remained unchanged when BPTU was added at 0 and 4 hpi (Figure 4D), cell death of the primary rotavirus-infected cells was delayed compared to mock-treated cells (Figure 4E). When BPTU was added at 8 hpi, rotavirus infection proceeded similarly to mock-treated cells with primary infection taking place within 8 h (Figure 4A,B,D). Similar to mock-treated cells, cell death of these primary infected cells occurred at 4 h post-detection of UnaG, and similar kinetics and amounts of secondary infection forming spatially restricted colonies were observed (Figure 4A–F). To determine whether inhibiting P2Y1 by BPTU blocked the rotavirus replication and production of de novo infectious virus particles, MA104 cells were infected with rotavirus and were either mock-treated or treated with BPTU at 0, 4, or 8 hpi. Viruses containing supernatants were collected at 12 and 24 hpi and evaluated for de novo virus production by plaque assay. Results show that, when BPTU is added at 0 hpi and 4 hpi, BPTU reduced de novo rotavirus production, while treatment at 8 hpi produces a similar amount of de novo viruses as compared to mock-treated cells (Figure 4G,H).

These results suggest that the P2Y1-mediated calcium waves are mostly important for the first round of infection as treating cells past 8 hpi does not impact the efficacy of the secondary round of infection (Figure 4C) and does not impact its kinetics (Figure 4A,D–F). This could suggest that, when a larger amount of virus is released from the primary infected cells, the requirement for P2Y1 signaling is no longer needed. To test if P2Y1 is required for the primary infection at a high MOI, MA104 cells were infected with rotavirus at an MOI of 3. BPTU was added to the cells at the time of infection (0 hpi), 4 hpi, or 8 hpi. Rotavirus infection was following by live-cell microscopy and plaque assay to determine virus spread and de novo virus production. Results show that, even at a high MOI, BPTU still leads to a decrease in primary infection, colony formation, and de novo virus production when added at 0 hpi and 4 hpi (Appendix A). Similar to when cells are infected with a low MOI of rotavirus (Figure 4), BPTU added at 8 hpi resulted in an infection like mock-treated cells (Appendix A). Together, these findings suggest that signaling through P2Y1 is required for the primary round of infection and that primary infected cells, through calcium wave signaling, prime neighboring cells for the secondary infection, making them not susceptible to the inhibition of rotavirus infection by the BPTU P2Y1 inhibitor.

### 3.4. High MOI Increase the Kinetics of Primary Rotavirus Infection

Our previous results show that calcium signaling is important for primary virus infection and colony formation independent of the MOI used for the primary infection (Figure 4 and Appendix A). However, when evaluating the time for primary infection from an MOI of 0.003 (Figure 1, Figure 2, Figure 3 and Figure 4) or an MOI of 3 (Appendix A), we noticed that, when infection was initiated with a higher MOI, the kinetics of primary infection was increased. To determine how the MOI impact infection, MA104 cells were infected with wild-type SA11 rotavirus expressing UnaG at four MOIs (0.064, 0.64, 6.4, and 64). Infections and colony formation were followed using live-cell microscopy by imaging every 30 min for 24 h. Results show, as expected, that increasing the MOI increased the number of infected cells (Figure 5A,B). Interestingly, the time to detect UnaG was significantly longer when cells were infected with a low MOI of rotavirus compared to cells infected with a higher MOI (Figure 5C). The time to detect UnaG for infection using a high MOI (MOI 64) (Figure 5C) was similar to the time for the onset of secondary infection after death of the primary infected cells (Figure 1E), which correspond to when de novo infectious rotavirus particles are released (Figure 2A). These results suggest that the increased kinetics of secondary infection occurs because of a high MOI of the rotavirus released locally following the death of the primary infected cells.

## 4. Discussion

Live-cell imaging studies have demonstrated that rotavirus spreads in a spatially controlled manner, where a single infected cell dies and subsequently infects surrounding cells. Using fluorescence imaging with UnaG, we detected rotavirus infection starting at 8 h post-infection. The initially infected cell underwent lysis around 12 h post-infection, with de novo infection initiated in surrounding cells four hours later. Notably, this secondary infection occurred more rapidly than the initial infection. To rule out early virus release as the cause of accelerated spread, we confirmed that no virus was released prior to cell lysis. To determine the mode of viral transmission, we employed a trypsin addition and quenching assays. These experiments demonstrated that a released virus was exposed to the extracellular environment and required trypsin activation, thereby ruling out spread within sealed vesicles. Furthermore, we found that the increased rate of secondary infection was driven by the high viral load released during the primary infection.

Our findings contrast with previous work by Santiana et al. [12], which showed that rotavirus can be released through microvesicles in cholangiocytes. Santiana et al. used electron microscopy to confirm that multiple rotaviruses were packaged within large vesicles [12]. Notably, these viruses were already cleaved and activated within the vesicles. The authors also demonstrated that this was not a cell culture artifact by isolating rotavirus-containing vesicles from animal stool samples [12]. Interestingly, our research reveals that viral release mechanisms may vary between cell types as we can show that our MA104 cells lyse and release viruses into the surroundings. Whether these viruses are associated with membranes is unknown; however, they are accessible to the outside environment and are not sealed within vesicles. Additionally, we show that viruses released from the initial infection are not processed and require access to extracellular proteases for virus propagation. Interestingly, while Santiana et al. can see that pigs and mice secrete rotavirus in stools, the amount of free viruses in stools vs. virus in vesicles depended on the animal where the stool was collected [12]. Some animals shed almost exclusively packaged viruses, while others only shed free virus. This variability underscores the need for further investigation into the mechanisms of viral shedding.

Upon the rotavirus infection of cells, the viral protein NSP4 interacts with the ER releasing calcium and initiating a calcium wave in surrounding cells [14]. Rotaviruses have been shown to require the P2Y1 receptor to propagate this induced calcium signal to neighboring cells [16]. These calcium waves have been suggested to lead to membrane disruption, which could lead to the onset of diarrhea. We evaluated whether P2Y1 propagated calcium waves could be responsible for priming the second round of infection seen in our rotavirus-infected cells. Interestingly, we found that the addition of the P2Y1 antagonist BPTU blocked initial rotavirus infection; however, when BPTU was added at 8 h post-infection, it had no effect (Figure 4 and Appendix A). This suggests that the P2Y1 receptor plays a more important role in initial rotavirus infection rather than subsequent rounds of infection. Inhibition of the P2Y1 receptors was also found to decrease rotavirus infection and plaque size by other labs, suggesting that the P2Y1 receptor plays a key role in the initiation of rotavirus infection [16,19].

Overall, our work shows that rotavirus increases its rate of transmission through infecting with increased amounts of virus during subsequent rounds of infections. These high virus loads allow the viruses to rapidly initiate the infection and production of de novo viruses. Increasing its rate of spread is key for viruses as they race against the host antiviral responses. While we see that secondary rounds of infection are faster than the first, overtime, the host responses will need to catch up to efficiently eliminate the infection. These current studies open important questions that will need to be validated in more physiologically relevant models. In the current configuration, we have evaluated the simian rotavirus infection of monkey kidney cells. Rotavirus normally infects the intestinal tract. We have also confirmed that a similar model of infection also holds true for human intestinal epithelial cells, suggesting that this growth and spatially restricted spread is not only a property of MA104 cells. However, this does not account for the three-dimensional nature normally found in the intestinal tract. Further studies employing intestinal organoids will be needed to confirm these findings. Increasing our understanding of the race between host and virus will be key to determine the best methods to combat rotavirus infections.

## Figures and Tables

**Figure 1 cells-14-00313-f001:**
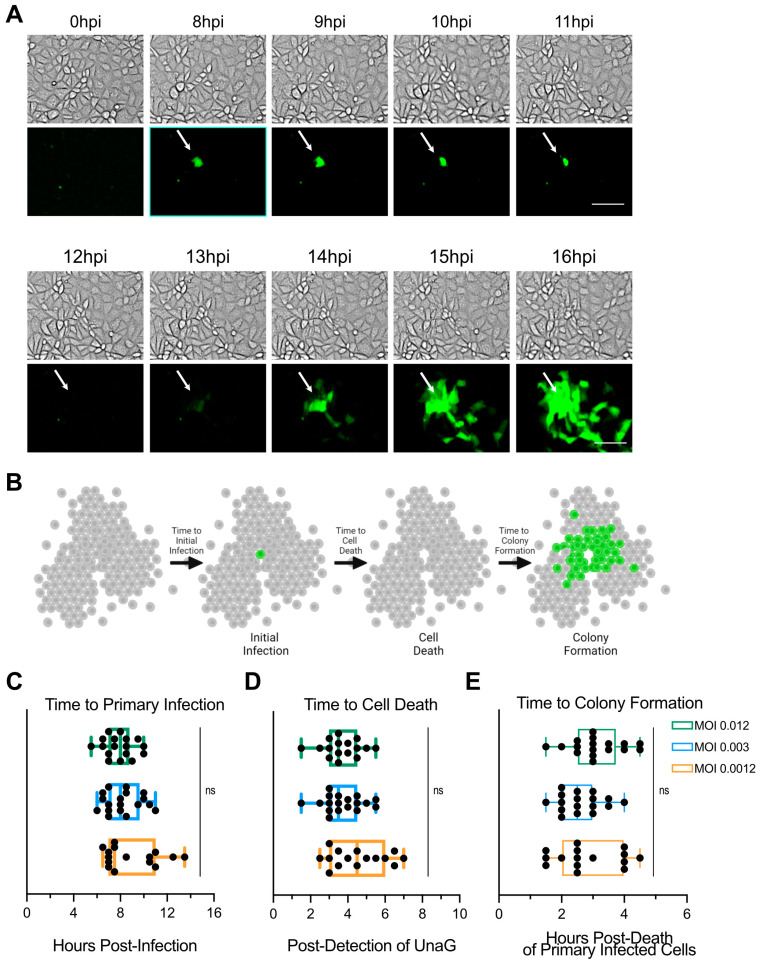
Rotavirus infection spreads in a spatially restricted manner. A confluent monolayer of MA104 cells is infected with WT rotavirus (RV) expressing UnaG at an MOI of 0.012, 0.003, or 0.0012. Virus infection is imaged using live-cell microscopy every 30 min for 16 h. (**A**) Representative brightfield (gray) and UnaG WT RV images (green). Scale bar = 100 μm. Arrows mark a primary infected cell through its lifetime as it dies and becomes a colony of infected cells. (**B**) The schematic of rotavirus infection and spread depicting primary infection, the cell death of primary infected cells and secondary infection forming a spatially restricted infected cellular colony surrounding the initial primary infected cells. (**C**) Quantification of the time to primary infection (time to detect UnaG). (**D**) Quantification of the time to cell death relative to the time to primary infection. (**E**) Quantification of the time to colony formation relative to the time of primary infected cell death. N = 15 fields of view from 3 independent experiments; and ns = not significant.

**Figure 2 cells-14-00313-f002:**
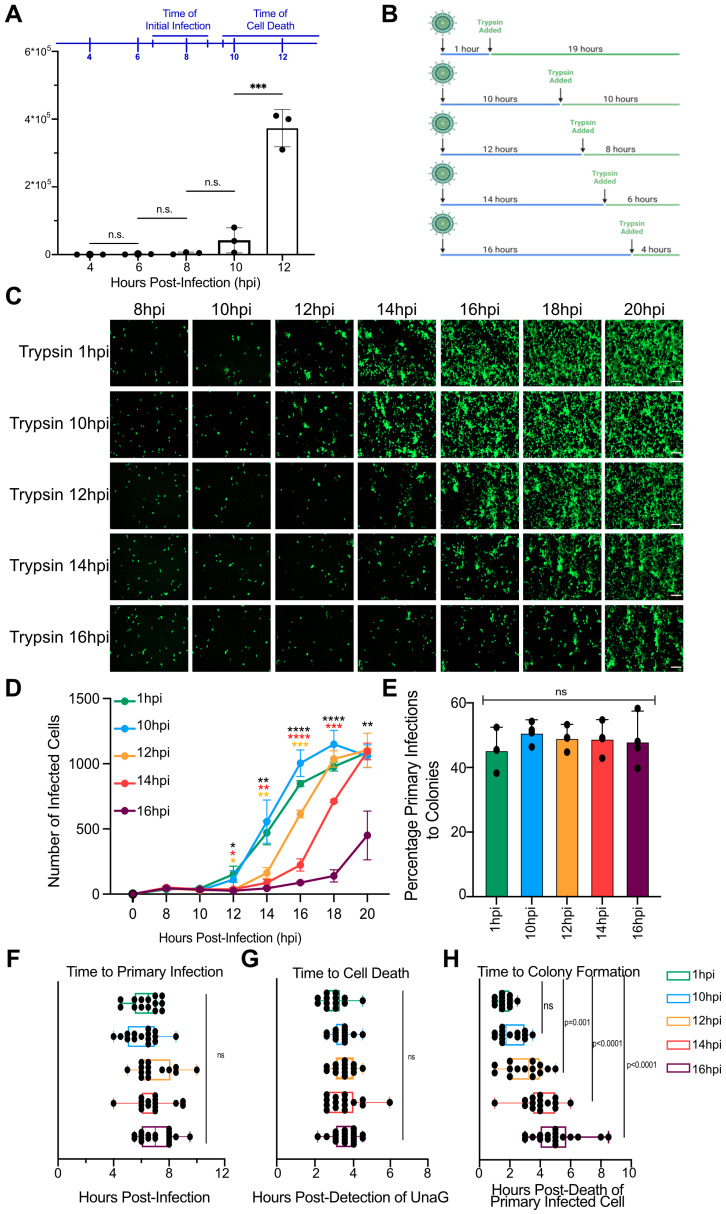
Rotaviruses are released and require trypsin activation for their spread: (**A**) A confluent monolayer of MA104 cells is infected with WT rotavirus (RV) expressing UnaG at an MOI of 0.003. Virus supernatants are collected at 4, 6, 8, 10, and 12 hpi. The production of de novo infectious rotavirus particles was analyzed by plaque assay. (**B**–**H**) A confluent monolayer of MA104 cells is infected with UnaG WT rotavirus (RV) at an MOI of 0.003. Trypsin is added at 1, 8, 10, 12, 14, 16, 18, or 20 hpi. Virus infection is imaged using live-cell microscopy every 30 min for 24 h. (**B**) Schematic showing the times of addition of trypsin following rotavirus infection. (**C**) Representative brightfield (gray) and UnaG WT RV images (green). Scale bar = 100 μm. (**D**) Quantification of the number of infected cells per field of view. (**E**) Quantification of the number of primary infected cells per field of view that leads to the formation of spatially restricted infected colonies (secondary infection of neighboring cells) at 16 hpi. (**F**) Quantification of the time to primary infection (time to detect UnaG). (**G**) Quantification of the time to cell death relative to the time to primary infection. (**H**) Quantification of time to colony formation relative to the time of primary infected cell death. N > 3, Statistics are performed by two-way ANNOVA. Scale bar represents standard deviation. * = *p* < 0.05, ** = *p* < 0.01, *** = *p* < 0.001, **** = *p* < 0.0001, ns = non-significant.

**Figure 3 cells-14-00313-f003:**
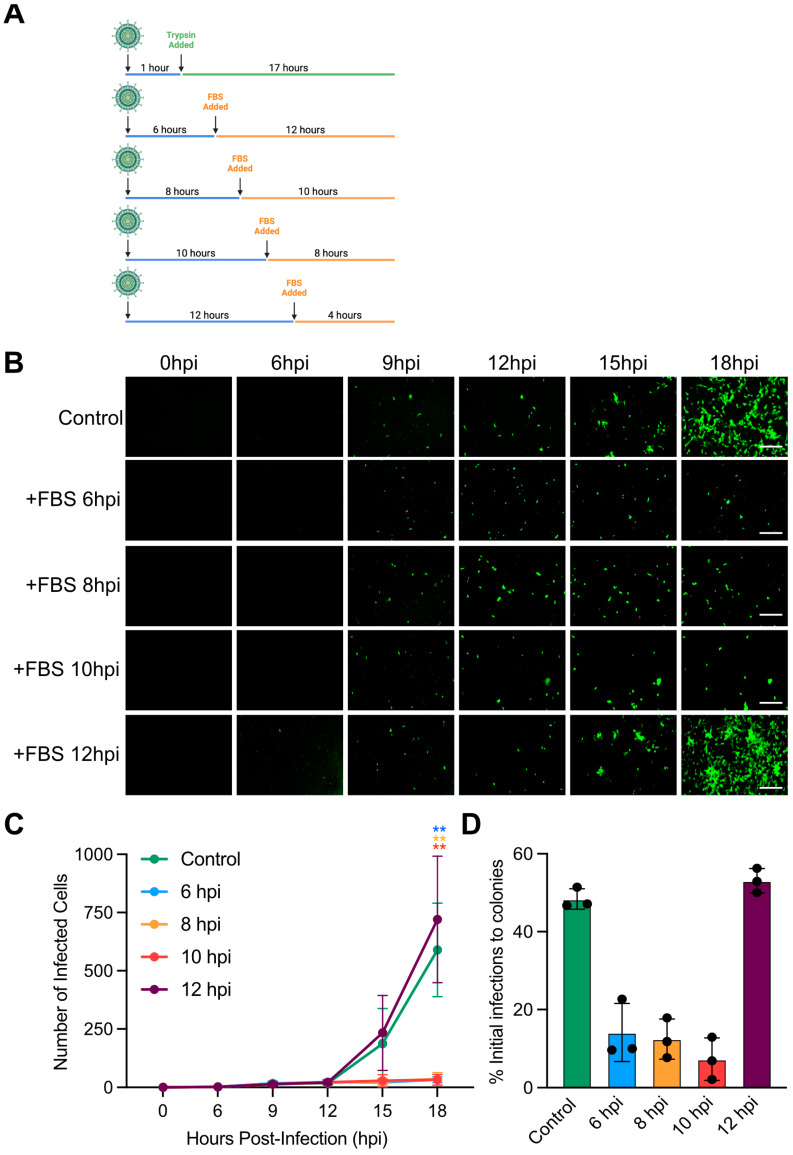
Trypsin neutralization impacts the formation of rotavirus-infected cellular colonies. A confluent monolayer of MA104 cells is infected with UnaG WT rotavirus (RV) at an MOI of 0.003. Trypsin is added at 1 hpi and is inactivated by the addition of FBS at 6, 8, 10, and 12 hpi. Virus infection is imaged using live-cell microscopy every 30 min for 16 h. (**A**) Schematic showing the time of addition of FBS following rotavirus infection. (**B**) Representative brightfield (gray) and UnaG WT RV images (green). Scale bar = 100 μm. (**C**) Quantification of the number of infected cells per field of view. (**D**) Quantification of the number of primary infected cells per field of view that leads to the formation of spatially restricted infected colonies (secondary infection of neighboring cells) at 16 hpi. N > 3, statistics are performed by two-way ANNOVA. Scale bar represents standard deviation. ** = *p* < 0.01.

**Figure 4 cells-14-00313-f004:**
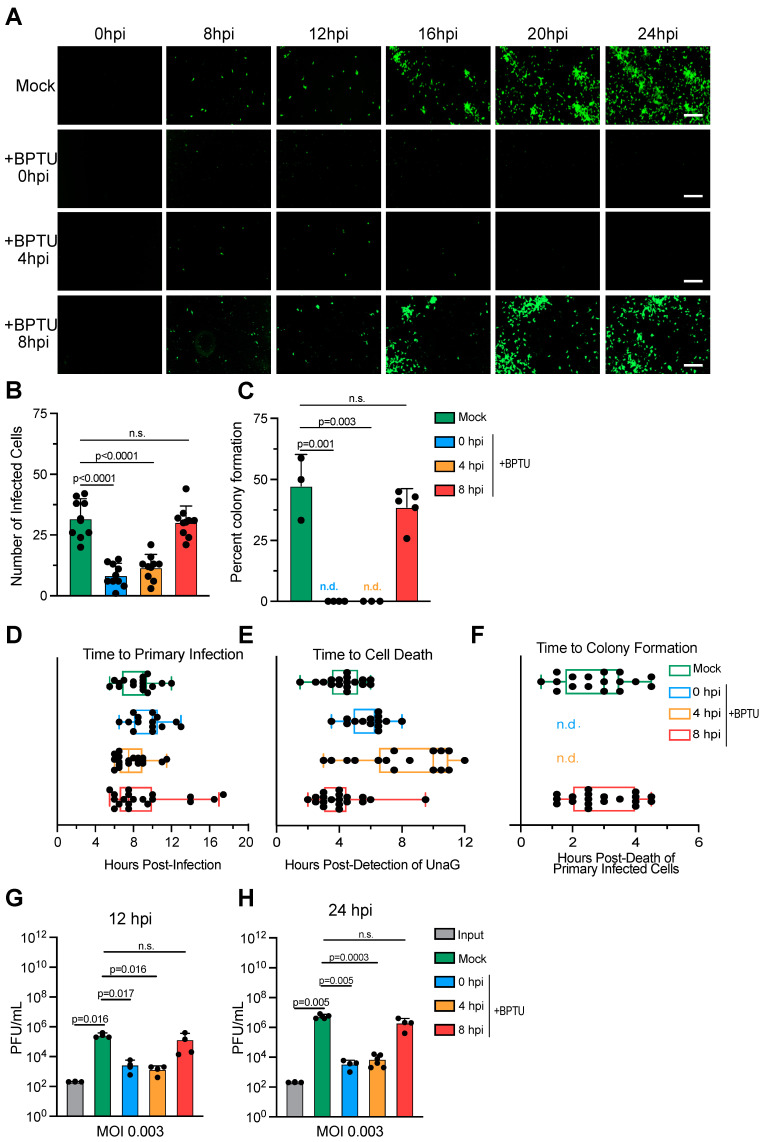
Blocking P2Y1 signaling reduces primary rotavirus infection and virus production. A confluent monolayer of MA104 cells is infected with UnaG WT rotavirus (RV) at an MOI of 0.003. Cells are either mock-treated with media +DMSO or treated with 10 μM of BPTU at 0, 4, and 8 hpi. Virus infection is imaged using live-cell microscopy every 30 min for 24 h. (**A**) Representative brightfield (gray) and UnaG WT RV images (green). Scale bar = 100 μm. (**B**) Quantification of the number of infected cells per field of view at 24 hpi. (**C**) Quantification of the number of primary infected cells per field of view that leads to the formation of spatially restricted infected colonies (secondary infection of neighboring cells) at 16 hpi. (**D**) Quantification of the time to primary infection (time to detect UnaG). (**E**) Quantification of the time to cell death relative to the time to primary infection. (**F**) Quantification of time to colony formation relative to the time of primary infected cell death. (**G**) Supernatants from A are harvested at 12 hpi, and the production of de novo infectious rotavirus particles is analyzed by plaque assay. (**H**) Same as G except supernatants are harvested at 24 hpi. (**A**–**F**) N = 10–15 fields of view from three independent experiments. (**G**,**H**) N = 3. Statistics are performed by two-way ANNOVA. Scale bar represents standard deviation. ns = non-significant.

**Figure 5 cells-14-00313-f005:**
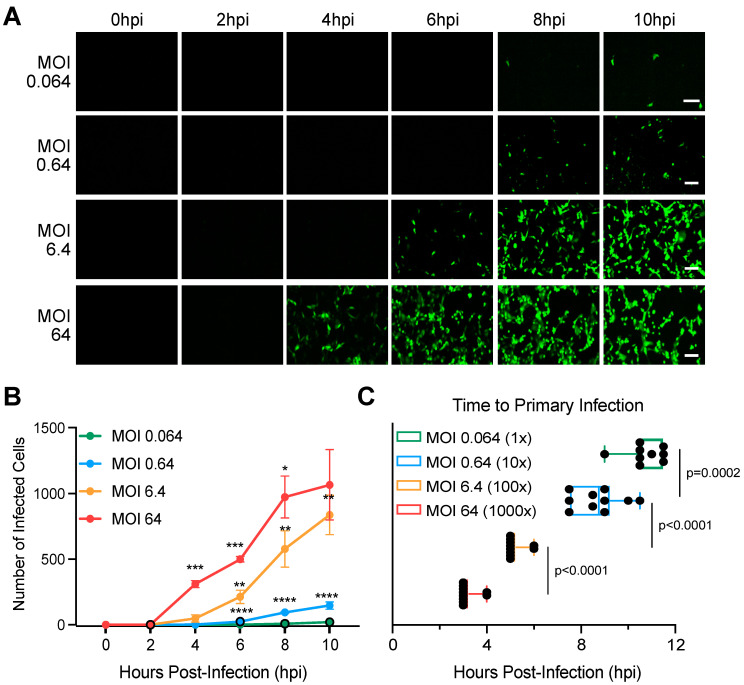
High MOI increases the kinetics of primary rotavirus infection. A confluent monolayer of MA104 cells is infected with UnaG WT rotavirus (RV) at an MOI of 0.064, 0.64, 6.4, and 64. Virus infection is imaged using live-cell microscopy every 30 min for 24 h. (**A**) Representative brightfield (gray) and UnaG WT RV images (green). Scale bar = 100 μm. (**B**) Quantification of the number of infected cells per field of view. (**C**) Quantification of the time to primary infection (time to detect UnaG). N > 3. Statistics are performed by two-way ANNOVA. Scale bar represents standard deviation. * = *p* < 0.05, ** = *p* < 0.01, *** = *p* < 0.001, **** = *p* < 0.0001.

## Data Availability

The original contributions presented in this study are included in this article/Appendix A, and further inquiries can be directed to the corresponding authors.

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
