# Peer review of "Rotavirus Spreads in a Spatially Controlled Manner"

_cells, 2025, doi:10.3390/cells14040313_

Round 1

Reviewer 1 Report

Comments and Suggestions for Authors

This study by Passerelli et al. is a careful dissection of the timing and pattern of spread by rotavirus within a cell monolayer. The core of the study is the observation that fluorescent reporter genes are expressed more rapidly in the secondarily infected cells surrounding a primary infected cell than in the primary cell itself. The study investigates a role for trypsin activation of the viral spike as well as a requirement for the P2Y1 receptor. MOI titrations reveal that the primary infection event can occur with a more rapid kinetic by simply adding a much higher MOI of viruses. The data are well presented, and the study is interesting. I have only few points of critique related to some of the mechanistic conclusions in the paper, which are not fully convincing.

Major points

1.        The conclusion that “primary infected cells, through calcium wave signaling, prime neighboring cells for secondary infection” is not really supported by the data. It could simply be that there are fewer infected cells that do not productively produce enough infectious virions to instigate spread.

2.        The experiments are not directly investigating the vesicular packaging of virions, but rather whether the activation by trypsin is necessary. This could address pre-activated virions within vesicles or other types of cleavage-independent spread mechanisms.

3.        There should be some control that the BPTU drug is behaving as expected.

4.        Could the BPTU drug block the rapid kinetic at high MOI Infection? It didn’t seem to block the rapid kinetic of spreading infection so testing this may indicate whether the rapid kinetic of high MOI is similarly independent of BPTU inhibition.

Minor points

1.        Typo introducing BPTU as an agonist but it is an antagonist.

2.        What is the timepoint for quantitation in Fig. 4B?

3.        Proofread for typos such as “primary” and other grammatical errors

Reviewer 2 Report

Comments and Suggestions for Authors

This manuscript presents interesting biological data on the behaviour of rotavirus in cell culture. By application of live cell imaging, timing and spread of rotavirus infection is very elegantly monitored in the MA104 cell line. This method allows the authors to present data over time, which they presented very nicely in microscopy images and comprehensive figures with time scales such as Fig 1C. This work provides interesting insights into how rotavirus spreads in cell culture over time, whether or not the virus is released in microvesicles and the role of calcium signaling. Although the work described here impresses as sound for the model used, the limitation lies in the lack of relevance for rotavirus infections in the human setting.  This should be addressed. 

1. The abstract should mention specifically that this work is performed in cell culture

2. It should be clear from the Methods that the simian rotavirus strain has been used

3. Limitations of the study should be addressed in the Discussion section: the model is  a 2-dimensional mono-culture of cells, and infection is performed with a simian rotavirus. Therefore, the relevance for the in vivo situation is very low. Infection dynamics will most likely be different in a 3D environment with physiologically differentiated human cells such as intestinal organoids. This might be difficult to study, but should at least be mentioned as a possibility to gain physiological relevance. The study presented here is a good first insight, but further investigations should be done in more relevant models. 

4. It was not completely clear to me how the quantification of the number of infected cells was done, such as described in Fig 2D. Was it just done by counting? I suggest to mention this more clearly in the Methods

5. I noticed the referring to figures in the Discussion. My suggestion is to  describe the keypoints of the study here in a narrative way without referring to the figures. 

6. Unfortunately I could only open movie 2, so I don;t know if the rest of the movies add anything. 

Reviewer 3 Report

Comments and Suggestions for Authors

In this manuscript, the authors show that the spread of Rotavirus depends on the MOI and there is cooperativity in the amount of virus in cells to the time taken for progeny release. The latter was shown to be via cell death and not vesicular secretion of active virus. Whereas separation of supposed 1Ëš and 2Ëš infection, virus release and 2Ëš infection has been shown only based on kinetic correlations, supportive evidences are missing to confirm if this is the case. There are missing info on how viruses were amplified which governs how the bystander priming if any was done. I have provided some specific comments below;

  1. Specific infectivity of virus used for 1Ëš infection versus released virus (used for 2Ëš infection) needs to be clarified, i.e., fraction of non-infectious particle to genome equivalent in these two fractions. Also, can the virus used to cause 1Ëš infection sensitive to trypsin? There is almost no information on how the virus was amplified and used for 1Ëš infection, i.e., is it the infectious supernatant or purified virus via CsCl or sucrose gradient based methods? 
  2. Virus purification info is vital when showing that 1Ëš infection is susceptible to ADP levels and not 2Ëš infection as has been suggested before. If culture supernatant is used, ADP levels in it must be measured in it.
  3. The authors have not separated primary and secondary infected cells efficiently, i.e., what is the possibility that 2Ëš infected cells show a fraction of primary infected cells with delayed gene expression (heterogeneity)? Co-culture experiments will clarify this.
  4. Fig 3A is unclear. In place of FBS, trypsin is written at several instances. Show that trypsin is added to all samples, and then FBS chase is performed, except n control where only medium is supplemented? Was this medium exactly the same as FBS supplemented medium?
  5. From the kinetics it appears that “calcium wave signalling” is required for entry (as for viruses like Adeno). Hence, it is likely that in presence of P2Y1, receptor independent or vesiculated viruses are produced that makes them not susceptible to BPTU anymore. There is a missing control that vesiculated pre-activated virus is sensitive to ADP/BPTU/P2Y1?

Round 2

Reviewer 3 Report

Comments and Suggestions for Authors

The authors' response are satisfactory.